# Detection of asymptomatic *Leishmania* infection in blood donors at two blood banks in Ethiopia

**Rezika Mohammed[1,2], Roma Melkamu[2], Myrthe Pareyn[3], Said Abdellati[3], Tadfe Bogale[2], Asinakew Engidaw[2], Abiy Kinfu[4], Tibebu Girma[4], Johan van Griensven** [3]*

**1** Leishmaniasis Research and Treatment Centre, University of Gondar, Gondar, Ethiopia, **2** Department of Clinical Sciences, Institute of Tropical Medicine, Antwerp, Belgium, **3** Department of Internal Medicine, University of Gondar, Gondar, Ethiopia, **4** National blood Bank, Addis Ababa, Ethiopia

* jvangriensven@itg.be

**Data Availability Statement:** All relevant data are within the paper and its Supporting Information files.

## Abstract

Visceral leishmaniasis (VL) is a disease caused by *Leishmania* parasites. While predominantly transmitted by sandflies, cases of VL transmitted through blood transfusion have been reported, particularly in immunocompromised recipients. Although *Leishmania* parasites have been found in blood donors in some VL endemic areas, this has never been studied in East-Africa, where HIV prevalence is relatively high. We established the prevalence of asymptomatic *Leishmania* infection and associated socio-demographic factors among blood donors presenting at two blood bank sites (Metema and Gondar) in northwest Ethiopia between June and December 2020. Metema is located in a VL-endemic area; Gondar has historically been considered VL non-endemic but as an outbreak of VL has occurred around Gondar, it was defined as previously VL non-endemic. Blood samples were tested by the rK39 rapid diagnostic test (RDT), rK39 ELISA, direct agglutination test (DAT) and qPCR targeting kinetoplast DNA (kDNA). Asymptomatic infection was defined as positive by any of these tests in a healthy person. A total of 426 voluntary blood donors were included. The median age was 22 years (IQR, 19–28 years); 59% were male and 81% resided in urban areas. Only one participant had a history of VL and three had a family history of VL. Asymptomatic infection was detected in 15.0% (n = 32/213) in Metema and 4.2% (n = 9/213) in Gondar. The rK39 ELISA was positive in 5.4% (n = 23/426), the rK39 RDT in 2.6% (11/426), PCR in 2.6% (11/420) and DAT in 0.5% (2/426). There were six individuals with two positive tests: one positive on rK39 RDT and PCR and five positive on rK39 RDT and ELISA. The prevalence of asymptomatic infection was higher in Metema (VL-endemic) and males but was not associated with age, a history of VL amongst family members or living in a rural area. Antibodies against *Leishmania* and parasite DNA was detected in a substantial number of blood donors. Future research should be directed at better defining the risk to recipients, including parasite viability studies and longitudinal studies amongst recipients.

**Funding:** The authors received no specific funding for this work.

**Competing interests:** The authors have declared that no competing interests exist.

## Author summary

Visceral leishmaniasis (VL) is a disease caused by *Leishmania* parasites. While predominantly transmitted by sandflies, cases of VL transmitted through blood transfusion have been reported, particularly in immunocompromised recipients. Whereas *Leishmania* parasites have been found in blood donors in VL endemic areas across the globe, this has never been studied in East-Africa. We studied how common *Leishmania* infections occur in blood donors presenting at two blood bank sites (Metema and Gondar) in northwest Ethiopia. Metema is located in a VL-endemic area; Gondar has historically been considered VL non-endemic but as an outbreak of VL has occurred around Gondar, it was defined as previously VL non-endemic. Blood samples were tested for the presence of the parasite (DNA) or antibodies against the parasite. Asymptomatic infection was defined as positivity on any of these tests in a healthy person. A total of 426 voluntary blood donors were included, predominantly young men living in urban areas. DNA or antibodies against Leishmania was detected in 15.0% in Metema and 4.2% in Gondar. There were six individuals with two positive tests: one positive on rK39 and PCR and five positive on rK39 and ELISA. The prevalence of asymptomatic infection was higher in Metema and males but was not associated with age, a history of VL amongst family members or living in a rural area. Antibodies against *Leishmania* and parasite DNA was detected in a substantial number of blood donors. Future research should be directed at better defining the risk to recipients, including parasite viability studies and longitudinal studies amongst recipients.

## Introduction

Visceral leishmaniasis (VL) is caused by parasites belonging to the *Leishmania donovani* complex. East-Africa currently carries the highest VL burden, followed by the Indian subcontinent and Latin-America. In East Africa, Ethiopia, South-Sudan and Sudan report the majority of VL cases. In Ethiopia, around 2500 to 4000 new cases of VL are thought to occur within the country per year [1–3].

Most individuals remain asymptomatic after infection, displaying direct (parasite detection with e.g. PCR) or indirect (e.g. serological tests) markers of infection but without symptoms. Some progress to overt disease (VL), manifesting with persistent fever, pancytopenia and hepatosplenomegaly. Without treatment, VL is fatal [4]. The risk of progressing to VL after infection is dramatically increased in individuals with concurrent HIV infection, which remains common in Ethiopia [5,6]. The north of Ethiopia has the highest rates of VL/HIV co-infection globally, with around 20% of VL cases found in HIV patients [5]. VL/HIV infection is associated with a poor prognosis, including poor treatment response and high rates of relapse and death [5].

The main route of transmission of *L. donovani* is from human to human by the bite of female phlebotomine sand flies. However, other modes of *Leishmania* transmission have been reported including mother-to-child (congenital VL), via needle-prick accidents and through sharing of infected needles in drug users [4,7,8]. There are also concerns that *Leishmania* could be transmitted via blood transfusion. Transmission of VL via blood transfusion has been demonstrated in hamsters and dogs [9,10]. In some VL endemic areas, a substantial number of healthy donors have been found asymptomatically infected with *Leishmania* [11]. Recent studies demonstrated viable *Leishmania* parasites in blood products of asymptomatically infected donors [11–13], even after prolonged storage, and these could be a source of infection for

patients receiving these blood products. Finally, probable cases of VL infected through blood transfusion have been reported across the globe [11,14]. As a consequence, in certain VL endemic areas, *Leishmania* screening of blood donors has been recommended by some authors [11,14].

In Ethiopia and other East-African countries neither blood donors nor blood products are currently screened for *Leishmania*, as there has never been any study in sub-Saharan Africa looking into this. In this study, we established the prevalence of asymptomatic *Leishmania* infection and associated socio-demographic factors among blood donors at two blood bank sites in northwest Ethiopia, one in a VL endemic area and one in an area historically considered to be non-VL endemic but providing care to VL patients coming from neighboring VL-endemic areas.

## Methodology

### Ethics statement

This study was approved by the Institutional Review Board of the University of Gondar (Reference number V/P/RCS/05/2272/2020). Permission to conduct the study was granted by the National blood bank. Written informed consent was taken from each participant.

### Study setting

This study was conducted at two blood bank sites in northern Ethiopia, the University of Gondar hospital and Metema hospital. The university hospital in Gondar is situated at 725 km northwest of the capital Addis Ababa and is a referral hospital that serves more than 5 million people. Although Gondar city has historically been considered as not to be situated in a VL endemic area, the hospital receives patients from endemic areas in the surroundings and provides VL diagnostic and treatment services. People living in the Gondar area sometimes travel to VL endemic areas, for instance for seasonal farming work. Moreover, a VL outbreak was detected in 2005 not far from Gondar in a highland area considered non-endemic for VL, presumably related to residents returning to their villages after seasonal work in the VL endemic low-lands [15]. Additionally, based on anecdotical observations, there is a suspicion that VL currently could exist in the wider Gondar area as well. For this reason, this site was defined as a "previously VL non-endemic area", as based on the observations above, VL could have been emerged in the Gondar area as well. Metema hospital is bordering Sudan and is located at 897 km northwest of Addis Ababa and 197 km from Gondar. The hospital is located in a VL high-endemic area and provides diagnostic and treatment for VL patients.

Within both hospitals, there is a blood bank providing service to patients admitted in Gondar, Metema and surrounding hospitals. Routinely, the donated blood from both sites is screened for HIV, Hepatitis B and C virus, and *Treponema pallidum* at the Gondar hospital blood bank laboratory. Moreover, donors with any signs or symptoms of an active disease are deferred to donate blood (9).

### Study design, population and recruitment

A cross-sectional study was conducted to determine the prevalence of asymptomatic *Leishmania* infection among blood donors. Between June and December 2020, all blood donors above 18 years old were recruited consecutively at the two blood bank sites in Gondar and Metema.

## Sample size calculation

Since there were no data on asymptomatic infection among blood donors in Ethiopia and East Africa, we used a 15.6% infection rate found in Brazil [16] to calculate the sample size. A single population proportion formula was used: $n = (z)^2 p (1 - p) / d^2$; z = 1.96; level of confidence: 95%; p = estimated proportion: 15.6%; d = margin of error: 5%. After including a 5% non-response rate, 213 participants had to be recruited at each study site, reaching a total sample size of 426 participants. Using a prevalence of 10% as could be expected from some studies enrolling healthy individuals in Ethiopia [17–19], a precision of 4.1% would be obtained.

## Data collection

Socio-demographic data (including age, sex, residency area, previous history of VL treatment, family members with VL, site of donation) were collected from each patient with a pretested questionnaire (draft version pretested with mock participants and adapted as needed). Additionally, whole blood and serum samples were collected immediately after blood donation using the same needle. In Metema, blood and serum samples were stored at -20˚C until they were shipped to Gondar where they were stored at -80˚C until analysis. In Gondar, rK39 RDT testing was performed immediately on serum and the remaining blood and serum samples were stored until analysis in batch.

## Laboratory analysis

To test the participants for asymptomatic *Leishmania* infection, serum samples were systematically tested by the rK39 rapid diagnostic test (RDT), direct agglutination test (DAT), rk39 IgG ELISA and whole blood by qPCR targeting kinetoplast DNA (kDNA).

The rK39 RDT (IT-LEISH, BioRad, USA) is a rapid immunochromatographic test for the qualitative detection of anti-leishmanial antibodies using a dipstick coated with recombinant rK39. The rK39 RDT was done using serum, following the instructions of the manufacturer.

The DAT/VL test (Institute of Tropical Medicine Antwerp, Belgium) includes a freeze-dried suspension of trypsin-treated and stained culture-form promastigotes of *L. donovani*. DAT testing was done on serum as described before (10). A DAT titer $\geq$ 1/1600 was considered positive (for asymptomatic infection), as used before [20].

The Serion *Leishmania* IgG ELISA (Serion Diagnostics, Würzburg, Germany) is a quantitative immunoassay which detects IgG antibodies against *Leishmania* rK39 in serum samples and was performed in duplicate for each sample according to the manufacturer's instructions. Optical densities (ODs) with less than 20% variability between the readings were considered valid and the first result was used. If variability was higher, the duplicates were retested. Results were interpreted as follows: values <10U/mL were negative, 10–15 U/mL were borderline positive, and $\geq$15U/mL were positive.

For detection of *Leishmania* kDNA by qPCR, DNA was isolated from 300 μL whole blood using the Maxwell 16 LEV Blood DNA purification extraction kit (Promega, Leiden, The Netherlands) with the automated Maxwell 16 Instrument (AS1000, Promega). After adding 300 μL lysis buffer and 30 μL proteinase K to the blood sample, the suspension was incubated at 56˚C at 400 rpm for 20 minutes and loaded into the Maxwell device according to the manufacturer's instructions. After extraction the DNA samples were stored at -80˚C until further analysis. In each batch of 15 samples, one negative extraction control (NEC), solely lysis buffer and proteinase K, was included. *Leishmania* DNA was detected using a kDNA qPCR as described before [20]. The reaction was run on a Rotor-Gene Q instrument (Qiagen, Venlo, The Netherlands). In each run, two positive (*L. donovani* DNA) and negative (elution buffer and PCR-grade water) PCR controls and the NECs were included. Results were expressed in

cycle threshold (Ct) values. Higher Ct values correspond with lower parasite DNA loads. For all samples with Ct values above 35, a repeated extraction and PCR was done to confirm positivity. If the repeat test was positive again, the Ct of the first run was used. In case this result was not confirmed in the second run, the sample was called weak positive and Ct of the first run was used for the analyses.

## Statistical analysis

EpiData (version 2.2.3.187, EpiData Association, Odense, Denmark) was used for data entry and Stata 15 for analysis. Data were summarized using frequencies and proportions for binary/categorical variables and medians (interquartile ranges (IQR)) for continuous variables.

The main outcome for the study was the prevalence of asymptomatic infection. Asymptomatic infection was defined if at least one of the *Leishmania* tests was positive (rK39 RDT, rK39 ELISA, DAT or qPCR) in a healthy person [21]. Additionally, a more stringent definition was used, considering weak (PCR) or borderline (rK39 ELISA) positive test results as negative and using a cut-off of $\geq 1/3200$ for DAT. Unless explicitly mentioned otherwise, all data shown in the results relate to the former (less stringent) definition. The Chi Square test was used to compare the proportion of positive *Leishmania* tests between both study sites. Factors associated with asymptomatic *Leishmania* infection were determined using logistic regression. Factors with a P-value $< 0.05$ in univariate analysis were included in multivariate analysis. The strength of the association was expressed using odds ratios (OR). The kappa statistic was calculated using the "kap" command in Stata.

## Results

### Description of the study population

A total of 426 voluntary blood donors were included in this study, 213 from Metema hospital and 213 from Gondar university hospital. The median age was 22 years (IQR 19–28 years). The majority of the donors (253/426; 59%) were male and resided in urban areas (347/426; 81%). Only one participant had a history of VL and three participants had a family history of VL. In Gondar, participants were younger and included more females. While in Metema 78 (37%) donors came from rural areas, all but one came from urban areas in Gondar (Table 1).

### Prevalence of asymptomatic *Leishmania* infection

All samples were tested with the different assays, except six samples from Metema that were not tested by PCR because the tubes were broken after storage. The prevalence of asymptomatic infection, defined as at least one positive *Leishmania* blood test was 15.0% in Metema and 4.2% in Gondar. The highest prevalence (31%) was seen in males aged 35 years and above from Metema. The most common positive tests were rK39 ELISA (n = 23; 5%) followed by rK39 RDT (n = 11; 3%) and PCR (n = 11; 3%). Only two (0.5%) samples were positive on DAT. The proportion of individuals positive on PCR and ELISA tests was significantly higher among blood donors in the VL endemic area of Metema (Table 2). The median Ct value for positive PCR results was 40.4 (39.6–40.6). Overlap of the different *Leishmania* tests used in the study was generally poor (Fig 1). The two patients with DAT positive results were seen at the Metema site, both with a titer of 1/1600. None of the two were positive for any other *Leishmania* test. Five ELISA positive and one qPCR positive donors were also positive by rK39 RDT (Fig 1). The correlation between the different tests is displayed in S1 Table. Using a more

**Table 1. Baseline characteristics of blood donors at two sites in northwest Ethiopia, 2020.**

| | Gondar hospital (N = 213) | Metema hospital (N = 213) |
|---|---|---|
| **Age (years); n (%)** | | |
| 18–24 | 156 (73) | 109 (51) |
| 25–34 | 40 (19) | 76 (36) |
| 35–45 | 9 (4) | 25 (12) |
| >45 | 8 (4) | 3 (1) |
| Median (IQR) | 22 (19–25) | 24 (19–30) |
| **Sex; n (%)** | | |
| Male | 108 (51) | 145 (68) |
| Female | 105 (49) | 68 (32) |
| **Residence; n (%)** | | |
| Urban | 212 (99.5) | 135 (63) |
| Rural | 1 (0.5) | 78 (37) |
| **Personal VL history; n (%)** | | |
| Yes | 1 (0.5) | 0 (0) |
| No | 212 (99.5) | 213 (100) |
| **Family member VL history; n (%)** | | |
| Yes | 2 (1) | 1 (0.5) |
| No | 211 (99) | 212 (99.5) |

IQR: interquartile range; VL: visceral leishmaniasis

**Table 2. Prevalence of asymptomatic *Leishmania* infection of blood donors at two sites in Northwest Ethiopia, 2020.**

| | Total | Gondar hospital | Metema hospital | P value[a] |
|---|---|---|---|---|
| rK39 RDT, n (%) | | | | 0.76 |
| Negative | 415 (97) | 208 (98) | 207 (97) | |
| Positive | 11 (3) | 5 (2) | 6 (3) | |
| rK39 ELISA, n (%) | | | | <0.001 |
| Negative | 403 (95) | 211 (99) | 192 (90) | |
| Positive | 23 (5) | 2 (1) | 21 (10) | |
| Titer > 15 U/mL | 16 | 2 | 14 | |
| Titer 10–15 U/mL | 7 | 0 | 7 | |
| DAT, n (%) | | | | 0.16 |
| Negative | 424 (99.5) | 213 (100) | 211 (99) | |
| Positive | 2 (0.5) | 0 (0) | 2 (1) | |
| PCR, n (%) | | | | 0.034 |
| Negative | 409 (98) | 211 (99) | 198 (96) | |
| Positive | 11 (3) | 2 (1) | 9 (4) | |
| Weakly positive | 4 | 1 | 3 | |
| Asymptomatic infection | | | | <0.001 |
| No | 385 (90) | 204 (96) | 181 (85) | |
| Yes | 41 (10) | 9 (4) | 32 (15) | |

DAT: direct agglutination test; ELISA: Enzyme-linked immunosorbent assay; PCR: polymerase chain reaction; RDT: rapid diagnostic test

[a] P-value based on Chi-squared test

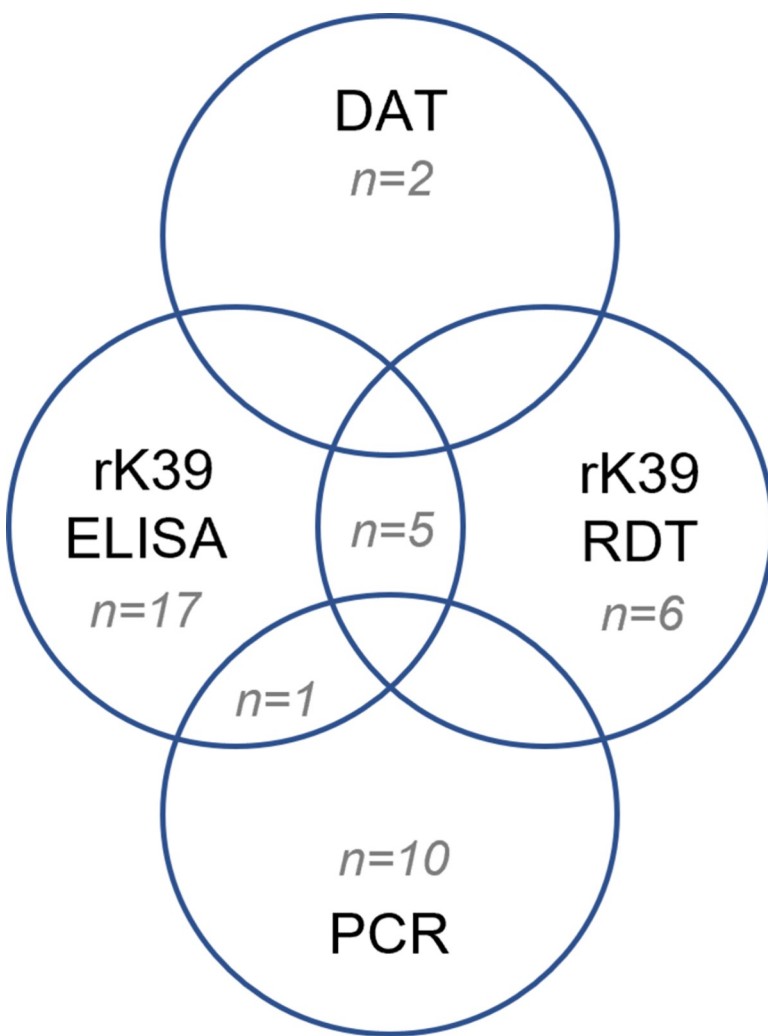

**Fig 1. Venn diagram with the results of the different tests to detect asymptomatic Leishmania infection.** DAT: direct agglutination test; ELISA: Enzyme-linked immunosorbent assay; PCR: polymerase chain reaction; RDT: rapid diagnostic test.

stringent definition of asymptomatic infection, the prevalence of asymptomatic *Leishmania* infection was 9.4% in Metema and 3.8% in Gondar.

Factors associated with asymptomatic Leishmania infection

In univariate analysis, a statistically insignificant association with asymptomatic infection was found for site of donation, with a higher risk of infection in the VL endemic site (Metema) and male sex. An increased risk was seen in donors between 35–45 years old, but the overall association between age and infection was not statistically significant (Table 3). In multivariate analysis, only the association with the site of the blood donation site remained statistically significant.

## Discussion

This is the first study in East-Africa describing the prevalence of asymptomatic *Leishmania* infection in (healthy) adult blood donors Asymptomatic *Leishmania* infection was found in 15.0% in Metema (the VL-endemic study site) and in 4.2% in Gondar (previously considered

**Table 3. Factors associated with asymptomatic *Leishmania* infection among blood donors at two sites in Northwest Ethiopia, 2020.**

| Variable | Asymptomatic infection; n/N (%) | Crude OR (95% CI) | P-value[a] | adjusted OR (95% CI) | P-value[a] |
|---|---|---|---|---|---|
| **Age, years** | | | 0.25 | | |
| 18–24 | 23/265 (9) | 1 | | | |
| 25–34 | 10/116 (9) | 1.0 (0.4–2.1) | | | |
| 35–45 | 7/34 (21) | 2.7 (1.1–6.9) | | | |
| >45 | 1/11 (9) | 1.0 (0.1–8.6) | | | |
| **Sex** | | | 0.030 | | 0.11 |
| Female | 10/173 (6) | 1 | | 1 | |
| Male | 31/253 (12) | 2.3 (1.1–4.8) | | 1.9 (0.9–4.0) | |
| **Residence** | | | 0.31 | | |
| Urban | 31/347 (9) | 1 | | | |
| Rural | 10/79 (13) | 1.5 (0.7–3.1) | | | |
| **Blood donation facility** | | | <0.01 | | 0.001 |
| VL endemic(Gondar) | 9 (4) | 1 | | 1 | |
| Previously VL non-endemic (Metema) | 31 (15) | 4 (1.9–8.6) | | 3.6 (1.7–7.9) | |
| **Family member with VL history** | | | 0.57 | | |
| No | 41/423 (10) | 1 | | | |
| Yes | 0/3 (0) | - | | | |

OR: odds ratio; VL: visceral leishmaniasis

[a] P-values based on logistic regression model

non-VL endemic). The prevalence was highest in male donors above 35 years old in Metema. While most cases only had positive serological tests, 11 also had detectable parasite DNA in their blood, albeit with high Ct values.

The prevalence of asymptomatic *Leishmania* infection in blood donors varies a lot between studies from different VL endemic areas around the world. While a recent systematic review calculated an overall prevalence of 11%, this varied widely across studies [22]. This large variability can to a large extent be explained by different VL-endemicity in the various regions, and by the type of tests used to determine asymptomatic *Leishmania* infection [21,22]. As to Ethiopia, asymptomatic *Leishmania* infection has been found in around 5–10% in the general population in VL-endemic areas [17–19], with the highest prevalence in young males. Similarly, VL disease is most commonly seen in young males, predominantly migrant workers [5].

Amongst the serological tests, rK39 ELISA had the highest yield, followed by the rK39 RDT. DAT had the lowest yield. Similar observations have been made in other studies [20]. While for East-Africa, DAT has been found more sensitive compared to the rK39 RDT to detect VL [23], this seems to indicate that this cannot automatically be extrapolated to detection of asymptomatic *Leishmania* infection. One key difference between asymptomatic and symptomatic (VL) *Leishmania* infection, is that in the former generally much lower levels of antibodies can be expected [24]. Possibly, during asymptomatic infections with lower levels of antibodies being produced, the better performance of the rK39 RDT ELISA compared to DAT could relate to higher immunogenicity of the rK39 antigen and/or longer persistence of anti-rK39 antibodies. The correlation between the tests was also generally poor, as also observed in other studies [25]. While on the one hand this would support the recommendation to combine different tests to increase the sensitivity to detect asymptomatic *Leishmania* infection, it could also carry a risk of decreasing specificity [21].

While the issue of screening blood donors for asymptomatic *Leishmania* infection has been discussed in various VL-endemic regions [11,14,26,27], further reflections are needed how to

address this in the context of Ethiopia and other East-African countries. The major question to be answered is whether blood donors in VL endemic regions should be systematically screened for asymptomatic *Leishmania* infection, and if found positive, what to do with the donated blood.

A basic limitation of all studies on asymptomatic *Leishmania* infection in general relates to the lack of a gold standard, and most studies have defined asymptomatic *Leishmania* infection as positivity on any of the *Leishmania* tests used [21,28]. The presence of *Leishmania* antibodies in the blood are an indirect marker for asymptomatic *Leishmania* infection and could merely be an indication of a past infection, which is now fully controlled or cleared, or could be due to cross-reactivity. Detection of parasite DNA in the blood at least indicates ongoing infection, as it has been shown that the DNA detected by PCR in the blood stems from viable parasites, and is degraded rapidly in the blood [29].

It is thus currently unclear at the global level which tests should be used to rule out asymptomatic *Leishmania* infection in blood donors [21]. Moreover, systematic screening for asymptomatic *Leishmania* infection would entail substantial additional costs and particularly when using molecular or complex serological testing, could come with additional delays. Molecular tests also have very limited availability in most VL-endemic regions in East-Africa.

Another question relates to what to do when a *Leishmania* infection is found in a donor, as in areas such as Metema with scarcity of blood in blood banks, this entailed 15% of all donors. While several case reports have been published of possible cases of transfusion-transmitted VL, this risk is currently not well defined [11,14,30]. In some areas, the risk is thought to be very minimal, but this has also been explained due to the systematic implementation of risk reduction practices such as leucofiltration or pathogen reduction techniques [11,14]. These are however currently not or hardly implemented in East African VL-endemic countries. Nevertheless, investing in these technologies has the potential to prevent transmission of a wide range of transfusion-related infectious pathogens. We also do not know well whether and how the risk of transmitting *Leishmania* infection relates to the different *Leishmania* markers in the blood donor, whereby several authors have suggested the risk to be higher if the parasite is detected directly (such as by PCR), compared to the mere detection of antibodies [31,32]. We note that in our study only 11 individuals had parasite DNA detected in their blood, and generally at very low levels.

On the other hand, the risk of developing VL after transfusion of contaminated blood is thought to be higher in immunosuppressed individuals such as patients living with HIV [11,14]. The HIV prevalence is fairly high in several of the VL endemic East African countries [5]. This raises the issue that at least for such patients, *Leishmania* infection should be ruled out in blood donors. In a recent Brazilian study, six of the 13 patients receiving infected blood seroconverted within six months after transfusion, but none developed VL during this time [16]. This could however be different in immunosuppressed individuals, or with longer follow-up.

As a way forward, studies assessing the viability and risk of infection of *L. donovani* parasites in blood products under different storage conditions would be useful. In parallel, longitudinal studies in recipients having received blood from donors with asymptomatic *Leishmania* infection would be valuable to assess the risk of VL, as has been done in Brazil.

While future research should be prioritized to the historical VL endemic areas, further studies should also assess the prevalence of asymptomatic infection in areas close to historic VL endemic areas, as in Gondar we still detected asymptomatic infection in 4% of the blood donors. To what extent these are true *Leishmania* infections or also include false-positives remains to be defined. On the other hand, this could also be an indication that VL is extending to new areas, beyond the low-lands, possibly related to global warming [33,34]. As mentioned above, a VL outbreak was detected around Gondar in 2005 [15].

One of the strengths of the study is that we had access to several *Leishmania* tests, including molecular tests, in a high-quality clinical research laboratory. Limitations include the lack of a gold standard to define asymptomatic *Leishmania* infection, similarly to all other comparable studies. PCR Ct values were high and some serological tests included positives with low titers, questioning their significance. We also did not collect detailed clinical and socio-demographic data, which could have helped to better define risk factors for asymptomatic *Leishmania* infection in blood donors. We enrolled the same number of individuals in each site. Given differences in prevalence of asymptomatic *Leishmania* infection between these sites, a better approach would have been to calculate sample size separately for each site, taking into account the lower prevalence of asymptomatic infection expected in the previously VL non-endemic site. The inclusion of blood donors from more endemic and non-endemic areas, and in larger sample size, would have been useful as well.

In conclusion, this is the first study documenting the prevalence of asymptomatic *Leishmania* infection in blood donors in a VL endemic area in East-Africa. The prevalence was 15% in the VL-endemic site and 4% in the site close to a VL endemic area and previously considered VL non-endemic, and highest in males above 35 years. Asymptomatic *Leishmania* infection was most commonly detected by serological tests, and less so by molecular tests. More research is needed to define whether screening for *Leishmania* infection should be systematically implemented in VL-endemic areas in East-Africa and what to do with blood found to be contaminated with *Leishmania*. This should include parasite viability studies and longitudinal studies amongst recipients to better quantify the risk of transfusion-related transmission of VL.

## Supporting information

**S1 Table. Agreement and Kappa score for the different Leishmania tests.** DAT: direct agglutination test; ELISA: PCR: polymerase chain reaction; RDT: rapid diagnostic test (DOCX)

## Acknowledgments

We would like to thank all individuals included in the study and all the staff at the Leishmaniasis Research and Treatment Center in Gondar. We also would like to thank Bart Jacobs for statistical advice.

## Author Contributions

**Conceptualization:** Rezika Mohammed, Johan van Griensven.

**Data curation:** Rezika Mohammed, Said Abdellati, Tadfe Bogale, Asinakew Engidaw, Abiy Kinfu.

**Formal analysis:** Rezika Mohammed, Roma Melkamu, Myrthe Pareyn, Said Abdellati, Abiy Kinfu, Tibebu Girma, Johan van Griensven.

**Methodology:** Roma Melkamu, Myrthe Pareyn, Said Abdellati, Tadfe Bogale, Asinakew Engidaw.

**Supervision:** Rezika Mohammed, Roma Melkamu, Myrthe Pareyn, Said Abdellati, Tadfe Bogale, Asinakew Engidaw, Tibebu Girma.

**Writing – original draft:** Rezika Mohammed, Myrthe Pareyn, Johan van Griensven.

**Writing – review & editing:** Rezika Mohammed, Roma Melkamu, Tadfe Bogale, Asinakew Engidaw, Abiy Kinfu, Tibebu Girma.

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
