## [Decision Letter · Decision Letter 0]

3 Aug 2022

Dear Dr. van Griensven,

Thank you very much for submitting your manuscript "Detection of asymptomatic Leishmania infection in blood donors at two blood banks in Ethiopia: should blood donors be systematically screened?" for consideration at PLOS Neglected Tropical Diseases. As with all papers reviewed by the journal, your manuscript was reviewed by members of the editorial board and by several independent reviewers. In light of the reviews (below this email), we would like to invite the resubmission of a significantly-revised version that takes into account the reviewers' comments. 

This paper is interesting because data on persons infected asymptomatically ba Leishmania parasites are very scarce. But the authors should ask advice for statistical analysis of the data as recommended by reviewer 2. Would transfer to PLoS One be an option for the authors?

We cannot make any decision about publication until we have seen the revised manuscript and your response to the reviewers' comments. Your revised manuscript is also likely to be sent to reviewers for further evaluation.

Sincerely,

Gabriele Schönian

Academic Editor

Charles Jaffe

Section Editor

This paper is interesting because data on persons infected asymptomatically ba Leishmania parasites are very scarce. But the authors should ask advice for statistical analysis of the data as recommended by reviewer 2. Would transfer to PLoS One be an option for the authors?

Reviewer's Responses to Questions

**Key Review Criteria Required for Acceptance?**

**Methods**

-Are the objectives of the study clearly articulated with a clear testable hypothesis stated?

-Is the study design appropriate to address the stated objectives?

-Is the population clearly described and appropriate for the hypothesis being tested?

-Is the sample size sufficient to ensure adequate power to address the hypothesis being tested?

-Were correct statistical analysis used to support conclusions?

-Are there concerns about ethical or regulatory requirements being met?

Reviewer #1: Methods are clearly described

Reviewer #2: -Are the objectives of the study clearly articulated with a clear testable hypothesis stated? YES

-Is the study design appropriate to address the stated objectives? YES

-Is the population clearly described and appropriate for the hypothesis being tested? YES

-Is the sample size sufficient to ensure adequate power to address the hypothesis being tested? Comment were provided in the review.

-Were correct statistical analysis used to support conclusions? Comment were provided in the review.

-Are there concerns about ethical or regulatory requirements being met? Comments were provided

Reviewer #3: Sample size calculation

Lines 132-136. 

I questioned the authors’ sample size calculation for two study sites, that is, recruiting an equal proportion of study participants given Metemma is one of the classical VL endemic areas in Ethiopia while Gonder city and surrounding areas are not yet recognized as one of VL endemic foci. In addition, the authors used a 15.6% infection rate reported in Brazil to estimate the number of study participants to be enrolled in this cross-sectional study. I strongly believe this is not appropriate given that published data (PMID: 33430946 plus the references 14-16 cited by authors) regarding the rate of asymptomatic VL in Ethiopian VL endemic foci are available. Blood donors usually represent apparently healthy individuals and are an appropriate target to study the rate of asymptomatic infection.

**Results**

-Does the analysis presented match the analysis plan?

-Are the results clearly and completely presented?

-Are the figures (Tables, Images) of sufficient quality for clarity?

Reviewer #1: Results are clearly described

Reviewer #2: -Does the analysis presented match the analysis plan? NO. See comments

-Are the results clearly and completely presented? No, see comments

-Are the figures (Tables, Images) of sufficient quality for clarity? NO (tables). See comments

Reviewer #3: Prevalence of asymptomatic infection

Lines 207-209

The authors of the present study found low seropositivity using DAT in comparison to the rK39 rapid test (RDT) and rK39 ELISA. Contrary to their findings, several studies [PMID: 28446246; PMID: 17942129] revealed the poor sensitivity of rK39-based serologic tests in Ethiopia which is partially related to the high genetic diversity of the L. donovani strains circulating in Ethiopia. On the hand, DAT is reported to be sensitive enough and has been recommended to be performed in those samples that were found to be negative by the rK39 test. Thus, the authors need to discuss the reason why they found relatively higher seropositivity by rK39 ELISA (5%) and rK39 RDT (3%) than by DAT (0.5%)?

**Conclusions**

-Are the conclusions supported by the data presented?

-Are the limitations of analysis clearly described?

-Do the authors discuss how these data can be helpful to advance our understanding of the topic under study?

-Is public health relevance addressed?

Reviewer #1: Conclusions are clearly described, and supported by the data

Reviewer #2: -Are the conclusions supported by the data presented? Yes

-Are the limitations of analysis clearly described? NO

-Do the authors discuss how these data can be helpful to advance our understanding of the topic under study? NO

-Is public health relevance addressed? YES

Reviewer #3: Discussion

Lines 266-273

Authors recommend the inclusion of the serologic rK39 RDT test in the systematic screening of blood donors in East African settings emphasizing targeting male donors as a logistically feasible approach. Implementing this would still be unlikely given (i) most blood donors in Ethiopia, if not all East African countries, are males plus (ii) the sensitivity and specificity of the recommended rK39 RDT in Ethiopia, including Metemma, are poor and may not be a reliable diagnostic test to rule out Leishmania infection. Additionally, antibody detection in blood and products from those donors may not necessarily indicate the potential to pose a sensible threat to blood recipients. Excluding positive male donors from donating blood on the basis of serologic tests, therefore, would result in major limitation on blood availability, thus causing more harm than benefit. Given the difficulty in screening-test selection, and testing cost as suggested by authors, looking for cost-effective Leishmania parasite inactivation and/or filtration methods using riboflavin and ultraviolet light and filtration leukodepletion filters, respectively of the blood and blood products derived from blood donors residing in VL endemic areas can be considered as effective and efficient alternatives to provide safe and immediate blood transfusion access to the needing blood recipients.

**Editorial and Data Presentation Modifications?**

Reviewer #1: (No Response)

Reviewer #2: See comments below

Reviewer #3: Specific comments

Title: 

Authors may need to change their manuscript title as their findings could not provide sufficient data for the inclusion of proper screening methods for screening blood donors.

I recommend “Prevalence of Leishmania donovani antibodies and DNA among blood donors at two blood banks in Northwest Ethiopia”.

In endemic areas, detection of antibodies might not be a surprise and signifies individuals’ exposure to the Leishmania donovani parasite. Those seropositive individuals can be (i) asymptomatic carriers and had spontaneously cured the parasite, (ii) asymptomatic individuals recently infected and not yet developed the disease, or (iii) individuals who had developed the disease and got cured following anti-leishmanial treatment (according to Porlino et al, 55% of treated VL patients were found seropositive). The latter is the case in which one of the participants in the present study had a VL history. Thus, detection of Leishmania may not be an appropriate phrase. This is further corroborated by studies that revealed the persistence of antibodies against rK39 in asymptomatic infections or previously infected patients for many months or years after the parasites have been cleared. These finds warrant the inaccuracy of using these tests to detect active infection. Further, the authors of this manuscript raised their concern in utilizing serodiagnosis given seropositivity may represent false positive cases due to cross-reactivity of antibodies with other pathogens in their discussion.

Authors affiliations:

The affiliations of some authors are not correct. For example, Johan van Griensven’s affiliation is not the Department of Internal Medicine, University of Gondar, Gondar, Ethiopia.

Abstract:

Lines 44-46. Please rephrase those sentences. As I pointed out above, detection of Leishmania is not appropriate it implies the detection of the Leishmania parasite or its component (e.g., amastigote, Leishmania specific antigen, or DNA). Detection of antibodies is a piece of indirect evidence regarding recent or past infections and may not necessarily indicate acute infection as described above. The authors’ recommendations also need to tone down as they are very challenging to implement in Ethiopian settings. Studies in Ethiopia, including areas where the present study was conducted, show that the majority (over 80%) of blood donors are males. In line with this, even systematic screening of male donors is logistically challenging.

Summary:

Lines 58-61. I recommend those detailed results should be moved to the abstract rather than appearing in the summary. 

Line 61. “There were six patients …………………. Authors should avoid calling seropositive or DNA-positive individuals as patients. According to Transfusion Guidelines, blood donors are required to be apparently healthy. Therefore, those six positive donors in the present study should be considered as hypothetical asymptomatic VL cases.

**Summary and General Comments**

Reviewer #1: This study fills an important knowledge gap on prevalence of asymptomatic infection among blood donors in and near VL endemic areas in Ethiopia. Although findings are straight forward, the interpretation, and especially the possible implications for recommendations regarding routine screening of donor blood, remains difficult, and is appropriately discussed.

Comments:

- The reference numbers in the text do not correspond with the correct references in the References section.

- Line 36 and 179: Asymptomatic infection is defined a positivity of any test. I suggest to add "in a health person".

- Line 72: the estimated annual VL incidence range of 3700 to 7400 is based on outdated estimations for underreporting from the early 00's of 2-4 times. WHO has published more recent estimations for underreporting of VL incidence in Ethiopia of 1.2 - 1.8 (WHO Weekly Epidemiological Record, No 22, 2016, 91, 287-296).

- Line 272: Authors suggest the possibility of screening of donor blood with rK39 RDT in the resource-limited East African setting. This seems a bit random and opportunistic, considering that the rK39 RDT only identified a small proportion of all serological positive cases, and the fact that no parasite DNA was found in any of the rK39 RDT positive cases. What does a positive (or negative) rK39 RDT then tell us?

- The overlap of positivity for the different tests is very low (no overlap between DAT and rK39, and only one between rK39 ELISA and PCR). This discrepancy between test results raises questions about the performance of the different tests to assess asymptomatic infection in healthy persons. The serological tests have been validated for diagnosis of infection in suspect symptomatic cases, who generally have high antibody titers, but not for confirming infection in asymptomatic persons with low antibody titers. I would like to see a discussion on the interpretation of the discrepancy of positivity results between the different tests.

Reviewer #2: Abstract

Page 3, line 36: correct to ' as positive by any of...'

Introduction:

Page 5, line 70: Correct ' In East Africa...' linguistically.

 71: are these the total number of cases or incidence rate per 100,000. Correct!

Page 5, line 76: What is meant by 'others'? Is it immunocompromised? Mention it/

Methodology

Page 7, line 106: better to mention the reference number of IRB approval

Page 7, line 123: you may use 'deferred' a common blood bank word used in such situations.

Page 8, line 134: Mention the name of the formula used in sample size calculation or cite a reference.

Page 8, line 139: what is a pretested questionnaire?

Page 8, line 141: was the whole blood EDTA? What is meant by the same needle? Do you mean the same phlebotomy/withdraw then distributed to different tubes?

Page 9, line 150: what is the rationale behind using three serological tests?

Page 9, line 161: correct to 'optical density, OD'

162" correct to ' 20% variability between the two reading..' or similar

Page 9, line 165: Did you use 300ul of whole blood or buffy coat, where we expect amastigotes to be?

Results

Table 1: re-edit the table according to journal style.

There is no need to include the P-value as it had no meaning here. The 2x2 table should give difference between VL and non-VL versus the exposure or variable like sex, male female.

Page 12, Line 208: in a real laboratory protocol, one test should be used, which of the four is recommended?

Page 13, line 214: what was the new stringent definition used?

Table 2: edit table. 

Is such studies the difference is established between diseased (VL) and non-diseased (non-VL) with suspected exposure/risk factor? It will not be useful to compare between the two hospitals.

Cross-sectional study are problematic for low prevalence diseases as the number of positive cases will be low for statistical inference, which is 23/426 in this case.

Page 14, line 227: Metema is VL-endemic area. Gondar is a VL non-endemic area. In light of this, the results are not surprising and expected. Look at the data differently as in mentioned in my comments above.

Table 3: see comments on table 2.

Discussion

Page 17, line 245: High among males above 35. Might be that males are the main donors compared females. This the age of donation in that area. This could be the explanation. Check!

Page 17. Line 259: Correct 'basis' to 'basic'.

Page 18, line 272: In this study rk39 ELISA detect the highest number of symptomatic VL cases. Thus should be adopted! or perhaps IFAT or ELISA as they are adopted for the WHO VL case definition.

Page 19, line 285: 'blood is scarce' do you mean blood donation?

Page 19, line 294: Correct 'DND' to 'DNA'.

Page 19, 295: rephrase the sentence 'On the other hand....'.

Page 20, 309: another limitation is the low number of VL (23/426) to draw statistical conclusion. This is a general problem in cross-sectional studies targeting low prevalence disease.

Figures

Fig 1: Good representation. Agreement can also be tested by kappa statistic which will make the figure stronger.

Reviewer #3: Given the high prevalence of immunocompromised individuals as the result of HIV/AIDS infection in Ethiopia and the limited or no data availability on the prevalence of asymptomatic L. donovani infection in Ethiopia, studies using highly sensitive and specific molecular assays over serological-based studies are of importance to evaluate the risk of Leishmania parasites infection in the blood donors. The present study by Mohammed et al study aimed to provide additional data on asymptomatic Leishmania donovani infection among Ethiopian blood donors residing in VL endemic and proxy regions to provide additional evidence for potential transmission L. donovani to blood transfusion recipients, particularly immunosuppressed individuals. However, like that of several similar studies that have been done in many VL endemic countries, this study had not proven the transmission of L. donovani by blood transfusion. To prove the possibility of L. donovani transmission via a blood transfusion to susceptible recipients, the authors could have employed a cohort study design that includes the follow-up of both blood donors and recipients for several years for the development of laboratory-confirmed clinical signs and symptoms compatible with VL using highly sensitive and specific molecular assays which are reliable for making decision active infection accurately. Additionally, a viability and infectivity test of L. donovani parasites in frozen blood products in blood banking procedures and storage conditions is required.

PLOS authors have the option to publish the peer review history of their article (what does this mean?). If published, this will include your full peer review and any attached files.

Reviewer #1: No

Reviewer #2: No

Reviewer #3: No
---

## [Decision Letter · Decision Letter 1]

27 Dec 2022

Dear Dr. van Griensven,

Thank you very much for submitting your manuscript "Detection of asymptomatic Leishmania infection in blood donors at two blood banks in Ethiopia" for consideration at PLOS Neglected Tropical Diseases. As with all papers reviewed by the journal, your manuscript was reviewed by members of the editorial board and by several independent reviewers. The reviewers appreciated the attention to an important topic. Based on the reviews, we are likely to accept this manuscript for publication, providing that you modify the manuscript according to the review recommendations. 

Sincerely,

Gabriele Schönian

Academic Editor

Charles Jaffe

Section Editor

Reviewer's Responses to Questions

**Key Review Criteria Required for Acceptance?**

**Methods**

-Are the objectives of the study clearly articulated with a clear testable hypothesis stated?

-Is the study design appropriate to address the stated objectives?

-Is the population clearly described and appropriate for the hypothesis being tested?

-Is the sample size sufficient to ensure adequate power to address the hypothesis being tested?

-Were correct statistical analysis used to support conclusions?

-Are there concerns about ethical or regulatory requirements being met?

Reviewer #2: The Authors have corrected all points of concern, however,

Odds ratio should be included in the 'statistics' section.

Reviewer #3: (No Response)

Reviewer #4: - Line 143: It is not clear why the rK39 RDT was not done at Metema immediately after sample collection as was done in Gondar. 

- The definitions for the main analysis and secondary analysis are not clear in the methods and the results sections. Perhaps, the manuscript will be clearer if these outcome analyses are defined and described in a separate subtitle. Are these definitions applicable only for DAT and PCR, or used in composite analysis with other test results too? The results section also does not clearly show the findings of main analysis and secondary analysis. For example: there are 2 DAT positive results (Table 2). Are these based on the main analysis (1:1600) or secondary analysis (1:3200)?

- The CT cut off value used to define the PCR result as positive or negative is not clear. It is stated that the analysis was repeated when the CT values were >35. On the other hand, the median CT value for the PCR positives was 40.4 (39.6 – 40.6) indicating majority had CT values >35.

**Results**

-Does the analysis presented match the analysis plan?

-Are the results clearly and completely presented?

-Are the figures (Tables, Images) of sufficient quality for clarity?

Reviewer #2: All were corrected, however

Tables 2 and 3: What is the statistical tool used to calculate the P-value. Mention it below the table.

 In both tables correct to ‘P-value’

Reviewer #3: (No Response)

Reviewer #4: see comments under methods

**Conclusions**

-Are the conclusions supported by the data presented?

-Are the limitations of analysis clearly described?

-Do the authors discuss how these data can be helpful to advance our understanding of the topic under study?

-Is public health relevance addressed?

Reviewer #2: OK

Reviewer #3: (No Response)

Reviewer #4: - Line 259-60: it is mentioned that lower levels of antibodies can be expected in asymptomatic infections. On the flip side, asymptomatic infection may actually have higher antibodies due to their strong immunity protecting them from developing the disease. In fact, the discrepancy between the DNA detection and antibody detection may indicate that those with positive test for antibody detection have cleared parasites than those who tested positive with PCR. These arguments need to discussed in parallel and supported by references if available.

**Editorial and Data Presentation Modifications?**

Reviewer #2: Minor modifications requested

Reviewer #3: (No Response)

Reviewer #4: Abstract

- Line 42: It does not make sense to analyze for association between history of VL with asymptomatic infection as there was only one person with history of VL in the data. 

Introduction

- Line 86: insert reference for the sentence, “Transmission of VL via blood transfusion has been demonstrated in hamsters and dogs”

**Summary and General Comments**

Reviewer #2: None

Reviewer #3: It would have been nice if authors had prepared a point-by-point response to each one of the reviewers’ comments, suggestions, and questions and showed the changes they made in their revised manuscript with a track of changes for fast evaluation whether they have correctly addressed each one of them. 

Nevertheless, authors have adequately addressed most of my prior comments, suggestions and questions. However, I have a couple of comments which authors need consider.

1. In principle, the prevalence of asymptomatic and symptomatic L. donovani infections is higher endemic areas than in non-endemic areas. Given this fact, using the same prevalence rate for estimation of the sample sizes for two study sites (Metema and Gondar) doesn’t sound statistically and scientifically. I still think the sample size calculation is misleading and a better explanation for using the same sample size for two study sites should be provided. If not, this has to be stated, in the revised manuscript, as one of the limitations of the study.

2. Authors pointed out the expansion of VL endemicity form historically VL endemic areas to previously nonendemic highland areas (e.g., Libo Kemkem) in Ethiopia possibly due to global warming and considered this as their additional reason for enrolling number of blood donors from Gondar. On the contrary, throughout their manuscript they keep saying nonendemic area. There is a suspect for occurrence local L. donovani transmission in Godar city and its surroundings due to documented hypothetical autochthonous cases that never had a travel history to the known VL endemic areas although it has not yet properly Investigated. I therefore kindly recommended authors to replace "VL nonendemic area" with” previously VL non-endemic area”.

3. The explanation that DAT has historically been developed for detection of symptomatic VL doesn’t sound. Because this explanation does also apply to rK39 ELISA as it also has originally developed and validated to detect symptomatic VL infections. In principle, DAT is meant to detect anti-leishmania antibodies. Similarly, rK39 ELISA also detects anti-leishmania antibodies. It is true that different levels of antibody response can be found in asymptomatic infections verses in symptomatic L. donovani infections- possibly higher level in VL patients and lower in individuals with asymptomatic L. donovani infection. The most plausible reason for relative higher sensitivity of rK39 ELISA as compared to DAT in detection of asymptomatic infection could be due to the high immunogenicity of the targeted antigen and the prolonged presentence of anti-rK39 antibodies. Discuss this.

Reviewer #4: This is a well written article in an area which is not well investigated. The researchers have come up with an important issue that needs attention. Such studies on transmission mechanisms of neglected infectious diseases will help in the overall control and elimination strategies. I believe that the authors have addressed previous reviewers' comments appropriately. I have a few comments see above under the different sections.

PLOS authors have the option to publish the peer review history of their article (what does this mean?). If published, this will include your full peer review and any attached files.

Reviewer #2: No

Reviewer #3: No

Reviewer #4: No

Figure Files:

Data Requirements:

Reproducibility:

References

---

## [Decision Letter · Decision Letter 2]

6 Feb 2023

Dear Dr. van Griensven,

We are pleased to inform you that your manuscript 'Detection of asymptomatic Leishmania infection in blood donors at two blood banks in Ethiopia' has been provisionally accepted for publication in PLOS Neglected Tropical Diseases.

Best regards,

Gabriele Schönian

Academic Editor

Charles Jaffe

Section Editor

Reviewer's Responses to Questions

**Key Review Criteria Required for Acceptance?**

**Methods**

-Are the objectives of the study clearly articulated with a clear testable hypothesis stated?

-Is the study design appropriate to address the stated objectives?

-Is the population clearly described and appropriate for the hypothesis being tested?

-Is the sample size sufficient to ensure adequate power to address the hypothesis being tested?

-Were correct statistical analysis used to support conclusions?

-Are there concerns about ethical or regulatory requirements being met?

Reviewer #2: See R1

Reviewer #3: (No Response)

**Results**

-Does the analysis presented match the analysis plan?

-Are the results clearly and completely presented?

-Are the figures (Tables, Images) of sufficient quality for clarity?

Reviewer #2: The author's have added the statistical tools below tables 2 and 3 and in the methodology (Statistical analysis) as requested.

Reviewer #3: (No Response)

**Conclusions**

-Are the conclusions supported by the data presented?

-Are the limitations of analysis clearly described?

-Do the authors discuss how these data can be helpful to advance our understanding of the topic under study?

-Is public health relevance addressed?

Reviewer #2: See R1

Reviewer #3: (No Response)

**Editorial and Data Presentation Modifications?**

Reviewer #2: From myside the manuscript is ready for publication (Accept).

Reviewer #3: (No Response)

**Summary and General Comments**

Reviewer #2: See R1

Reviewer #3: (No Response)

PLOS authors have the option to publish the peer review history of their article (what does this mean?). If published, this will include your full peer review and any attached files.

Reviewer #2: **Yes: **Prof. Amer Al-Jawabreh

Reviewer #3: **Yes: **Tesfaye Gelanew

---

## [Editor Report · Acceptance letter]

4 Mar 2023

Dear Dr. van Griensven,

We are delighted to inform you that your manuscript, "Detection of asymptomatic *Leishmania* infection in blood donors at two blood banks in Ethiopia," has been formally accepted for publication in PLOS Neglected Tropical Diseases.

Best regards,

Shaden Kamhawi

co-Editor-in-Chief

Paul Brindley

co-Editor-in-Chief
